



# Strong Light Absorption Induced by Aged Biomass Burning Black Carbon over the Southeastern Tibetan Plateau in Pre-monsoon Season

Tianyi Tan[1], Min Hu[1], Zhuofei Du[1], Gang Zhao[1], Dongjie Shang[1], Jing Zheng[1], Yanhong Qin[1], Mengren Li[1], Yusheng Wu[1], Limin Zeng[1], Song Guo[1], Zhijun Wu[1]

[1]State Key Joint Laboratory of Environmental Simulation and Pollution Control, International Joint Laboratory for Regional Pollution Control, Ministry of Education (IJRC), College of Environmental Sciences and Engineering, Peking University, Beijing, 100871, China

*Correspondence to*: Min Hu (minhu@pku.edu.cn)

**Abstract.** During the pre-monsoon season, biomass burning (BB) activities are intensive in southern Asia. Facilitated by westerly circulation, those BB plumes can be transported to the Tibetan Plateau (TP). Black carbon (BC), the main aerosol species in BB emissions, is an important climate warming agent, and its absorbing property strongly depends on its size distribution and mixing state. To elucidate the influence of those transported BB plumes on the TP, a field campaign was conducted on the southeast edge of the TP during the pre-monsoon season. It was found that the transported BB plumes substantially increased the number concentration of the atmospheric BC particles by 13 times, and greatly elevated the number fraction of thickly-coated BC from 52% up to 91%. Those transported BC particles had slightly larger core size and much thicker coatings than the background BC particles. However, the coating mass was not evenly distributed on BC particles with different sizes. The smaller BC cores were found to have larger shell/core ratios than the larger cores. Besides, the transported BB plumes strongly affected the vertical variation of the BC's abundance and mixing state, resulting in a higher concentration, larger number fraction and higher aging degree of BC particles in the upper atmosphere. Resulted from both increase of BC loading and aging degree, the transported BB plumes eventually enhanced the total light absorption by 15 times, in which 21% was contributed by the BC aging and 79 % was contributed from the increase of BC mass. Particularly, the light absorption enhancement induced by the aging process during long-range transport has far exceeded the background aerosol light absorption, which implicates a significant influence of BC aging on climate warming over the TP region.

## 1 Introduction

Biomass burning (BB) is an important source of atmospheric aerosols, exerting a significant impact on the regional and global climate system (von Schneidemesser et al., 2015;Jacobson, 2014). BB is widely occurring around the world, as the means of heating, cooking, farming, etc. In some regions, BB has been becoming a serious environmental issue, which is harmful for both the public health and air quality, for example, the raging wildfires in Australia and California, USA this year. Due to the climate warming, this kind of natural BB event may increase in prevalence and severity in the future (Westerling et al.,



2006;Abatzoglou and Williams, 2016). Black carbon (BC) is the main aerosol species in BB emissions. It can strongly absorb light across the whole solar spectrum. BC is an important global warming agent, with an estimated radiative forcing of +1.1 Wm-2 (only secondary to $CO_2$) (Bond et al., 2013).During the transport in the atmosphere, BC aerosols will experience complex aging process and gradually become internally mixed with other aerosol species (non-BC coatings), which will lead

to a great change of BC's properties (Jacobson, 2001). Those coatings on BC will enhance the light absorption of BC through the so-called lensing effect (Lack and Cappa, 2010), which will further enhance the refractive heating of BC (Jacobson, 2001;Peng et al., 2016). Quantifying the light absorption enhancement of BC requires the information of size distribution, mixing state, morphology and chemical compositions(Fierce et al., 2016;Fierce et al., 2020).

Tibetan Plateau (TP), known as the Third Pole and the roof of the world, is the highest plateau on the Earth and owns the

largest store of ice outside of the polar regions (Qiu, 2008;Ramanathan and Carmichael, 2008). The TP plays a vital role in the large-scale atmospheric circulation and the hydrological cycle of the entire Asia continent (Lau et al., 2010;Yao et al., 2019). The TP has long been considered as one of the remote regions that are relatively less influenced by the pollutants emitted from human activities. However, growing evidences prove that anthropogenic emissions from the surrounding settlements have influenced this pristine place (Li et al., 2016;Cong et al., 2013). The warming rate of the TP has been found higher than the

rate of global warming (Xu et al., 2009;Chen et al., 2015). Recent studies show that light-absorbing aerosols (e.g. black carbon and mineral dust) are partially responsible for the rapid warming and the accelerated glacier retreat over the TP (Qiu, 2008;Yao et al., 2019). The TP is surrounded by several global hotspots of BC emissions (Bond et al., 2013). Every year open biomass burning (BB) is prevailing in South and Southeast Asia during the pre-monsoon season, resulting in large amounts of BC emissions (Streets et al., 2003;Duncan et al., 2003;Cong et al., 2015b;Pani et al., 2020). With the facilitation of meteorological

condition during this period, the significant amounts of BC can be transported into the TP and become a major source of BC in Tibetan region (Zhao et al., 2013). Previous studies reveal that southern Asia contributes more than half amount of BC to the Tibetan region on annual basis and become the dominant source during pre-monsoon seasons (Kopacz et al., 2011;Li et al., 2016;Lu et al., 2012;Zhang et al., 2015).

Nevertheless, existing researches on BC aerosols over the TP are limited to the mass concentration measurement, and only a

few studies pay attention to BC's mixing state. By using the electron microscopy technics, researchers observed the morphology of single BC particles in the TP and found that most of sampled BC particles were internally mixed with sulfate and organic matters (Cong et al., 2009;Li et al., 2015;Yuan et al., 2019;Zhang et al., 2001). In addition, there are two studies quantifying the number fraction of thickly coated BC in northeastern TP and found a percentage of around 50% (Wang et al., 2015;Wang et al., 2014). Therefore, more accurate information on the BC mixing state especially the size-resolved mixing

state is still in need when estimating the BC's warming effects on the TP.

In this study, an intensive field campaign was conducted during the pre-monsoon season at a remote background site of the Tibetan Plateau. To characterize the mixing state of BC, a combined system of differential mobility analyzer and single-particle soot photometer (DMA-SP2) was applied to obtain a robust size-resolved coating thickness of BC-containing particles (BC particles for short). By examining the size distribution, mixing state, and light absorption properties of BC, this study revealed



the impact of transported aged BB plumes on the atmospheric environment over the TP, which can serve as constraints for climate models to help with improving our understanding how human activities affect the global climate change.

## 2 Methodology

### 2.1 Site description

The measurement station is located on Mt. Yulong (27.2°N, 100.2°E), with an altitude of 3410 m a.s.l. (Fig. 1). Mt. Yulong is
situated at the southeastern edge of the Tibetan Plateau (TP), a transition zone that connects the vast low-altitude areas in southern Asia to the high-altitude TP. Different from the south edge of TP, the gentle slope of southeastern TP provides a geographical convenience for the import of air pollutants from southern Asia to the TP. The measurement period lasted from 22 March to 4 April 2015, corresponding to the pre-monsoon season in the region. The meteorological condition is shown in Fig. S1. More detailed information about the measurement site can also be found in Zheng et al. (2017).

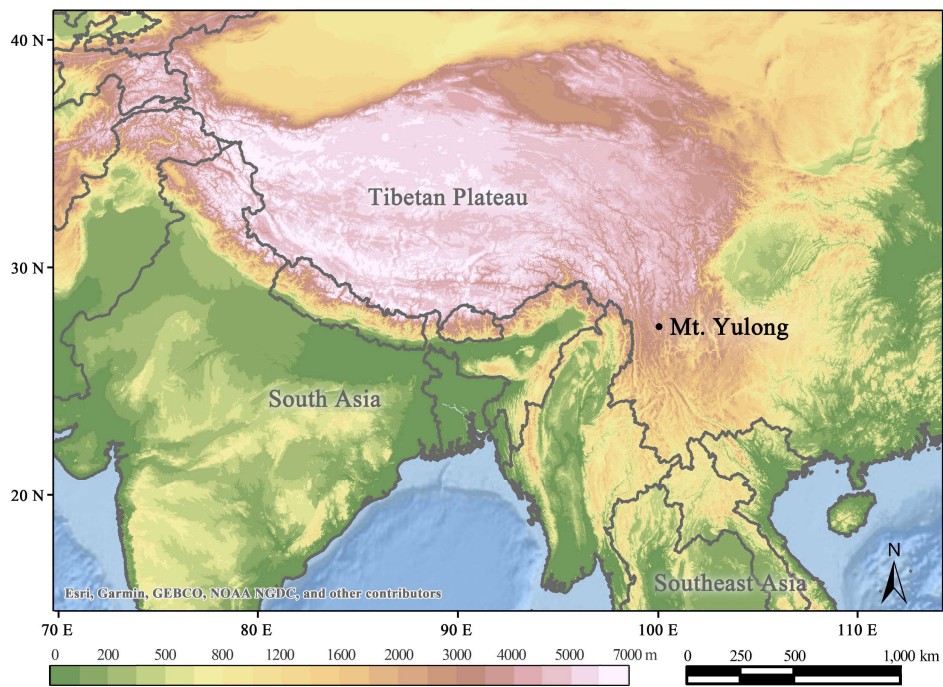


**Figure 1.** Location of measurement site at Mt. Yulong. Elevation data from SRTM 90m DEM Version 4, base map from ESRI. World Ocean Base, manipulating by ArcMap Version 10.2. Redlands, CA: ESRI, 2014.



## 2.2 Experimental Setup

During the field measurement, BC's properties were analysed by a DMA and SP2 combined system. After being dried by a
silica-gel diffusion dryer, the ambient particles were introduced into a differential mobility analyzer (DMA, Model 3081, TSI
Inc., USA), in which particles were separated according to their electrical mobility (mainly determined by the particle size).
Afterwards, size-resolved particles were driven into a single particle soot photometer (SP2, Droplet Measurement
Technologies, USA) and a paralleled condensation particle counter (CPC, Model 3022, TSI Inc., USA). In the SP2, the particles
passed through an intracavity laser beam (Nd: YAG, $\lambda$=1064 nm) separately. During this process, the scattering signals of each
particle were detected and recorded. If the particle contains black carbon (BC), it would be heated to its incandescence point
and the incandescence signals would also be recorded. The configurations of the DMA-SP2 coupled system were set as
following: the sheath flow rate of the DMA was 1.5 L/min and the aerosol flow rate was 0.45 L/min; the CPC consumed 0.3
L/min and the rest 0.15 L/min air was introduced into the SP2.

## 2.3 Data analysis

### 2.3.1 SP2 data

For non-absorbing particles, the peak intensity of the scattering signal is proportional to the scattering cross-section of the
particle, which mainly depends on the particle size. However, for absorbing particles (mainly BC-containing particles), the
scattering signal will be lost due to the particle evaporation (Gao et al., 2007;Moteki and Kondo, 2008). To obtain the full
scattering signal of these BC-containing particles, the LEO-fit method (Gao et al., 2007) was applied in this study and the
leading-edge signals were determined by using the method proposed by Liu et al. (2014) in their paper. Then, based on the
calibration of the scattering signal, the scattering cross-section of both absorbing and non-absorbing particles could be
determined. For BC-containing particles, the peak intensity of the incandescence signal is proportional to the BC mass in each
particle, which is unbiased by the mixing state of BC. The calibration of the incandescence signal was also conducted by using
the combined DMA-SP2 system, and Aquadag was used as the reference material. The formula provided by Gysel et al. (2011)
was applied to convert the mobility diameter into mass for Aquadag. Considering the different sensitivity of the SP2 response
to Aquadag and ambient BC, a correction of 0.75 was applied to derive the BC mass in the ambient particles (Moteki and
Kondo, 2010;Laborde et al., 2012).

### 2.3.2 Mixing state of black carbon

BC's Mixing state is referred to the state how BC is mixed with other particle-phase species. As the aging process proceeds,
the freshly emitted BC particles will be gradually coated by other chemical species and convert from bare (or thinly-coated)
BC into thickly-coated BC. To quantify BC's mixing state, there are three parameters that were used in this study: the number
fraction of thickly-coated BC, the shell/core ratio and the coating thickness. The separation of thinly- and thickly-coated BC
is based on the time delay between the occurrence of the peaks of the scattering signal and the incandescence signal (Moteki



and Kondo, 2007;Shiraiwa et al., 2007). Since this method does not require any assumptions or complex calculation, the
parameter of the thickly-coated fraction can be compared between different studies. The shell/core ratio denotes the diameter
ratio of the whole particle diameter ($D_p$) to the BC core dimeter ($D_c$), following Eq. (1). The coating thickness can be seen the
thickness of the shell, which equals to the half of the difference between $D_p$ and $D_c$ (Eq. 2).

$$\text{shell/core ratio} = \frac{D_p}{D_c}, \tag{1}$$

$$\text{coating thickness} = \frac{D_p - D_c}{2}, \tag{2}$$

In previous SP2 studies, the $D_p$ usually refers to the optically equivalent diameter of the whole particle diameter, which is
derived from the Mie calculation with several presumed input parameters (Taylor et al., 2015). Meanwhile, the $D_c$ refers to the
volume-equivalent diameter (or mass-equivalent diameter) of the BC core, by assuming a fixed density for a void-free spherical
BC core (1.8 g/cm$^3$ is widely used). However, the BC cores in ambient BC particles always contain some inside voids and
tend to contain more voids in thinly-coated BC particles. Thus, using a density of 1.8 g/cm$^3$ will underestimate the core size
and overestimate the coating thickness (and shell/core ratio) to some extent. In this study, with the benefit of the DMA-SP2
coupled system, the $D_p$ can be directly measured by the DMA, which is the mobility diameter of the whole particle. To derive
the $D_c$, instead of using the fixed density, we applied the closure study suggested by Zhang et al. (2018) to derive the size-
dependently effective density for thinly-coated BC particles (Fig. S2), and adopted a density of 1.2 g/cm$^3$ for thickly-coated
BC particles (more details on the calculation method can be found in Zhang et al.(2018)) .The multiply charged particles
induced by the DMA can be removed by comparing the mobility diameter with the optical diameter derived from the SP2 data.
Figure S3 shows the comparison between the mobility diameter and optical diameter of thinly- and thickly-coated BC for
single-charged particles. It should be noted here, when it comes to the BC size distribution, the mass-equivalent diameter of
BC cores (assuming a density of 1.8 g/cm$^3$) were adopted in this study for direct comparison with previous studies.

### 2.3.3 Light absorption of black carbon

Light absorption of coated BC particles was calculated based on Mie theory (Bohren and Huffman, 1983) . The refractive
index of BC coatings is the volume-average of different chemical species in BC's coating, with the assumption that the coatings
share the same chemical composition with the whole submicron aerosols. The chemical composition was measured by a high-
resolution time-of-flight aerosol mass spectrometer (AMS; Aerodyne Research Inc., Billerica, MA, USA). The density and
refractive index of different chemical species is from Table 1 of Barnard et al. (2010) and references therein.  For the
simplification of optical calculation, all the BC particles were sized into different bins according to their core diameters, and
all the BC particles in each bin were assumed to have the same shell/core ratio (median), which also means a united absorption
cross-section ($C_{abs,I}$, calculated from the Mie theory ) value in each bin. To calculate the bulk mass absorption cross-section
(MAC) for the particle population, the absorption cross-section of BC particles in each bin ($C_{abs,i}$) was summed and divided
by the sum of BC mass in each bin (Eq. 3).





$MAC = \frac{\sum_{i=1}^{n} C_{abs,i} \times N_i}{\sum_{i=1}^{n} M_{bc,i} \times N_i}$,    (3)

In Eq. (3), $n$ stands for the total number of size bins. $C_{abs,i}$, $M_{bc,i}$ is the absorption cross-section and BC mass of a particle in $i$th bin, respectively. $N_i$ is the total particle number in $i$th bin.

## 3 Results

### 3.1 Observational evidence for the transport of biomass burning black carbon

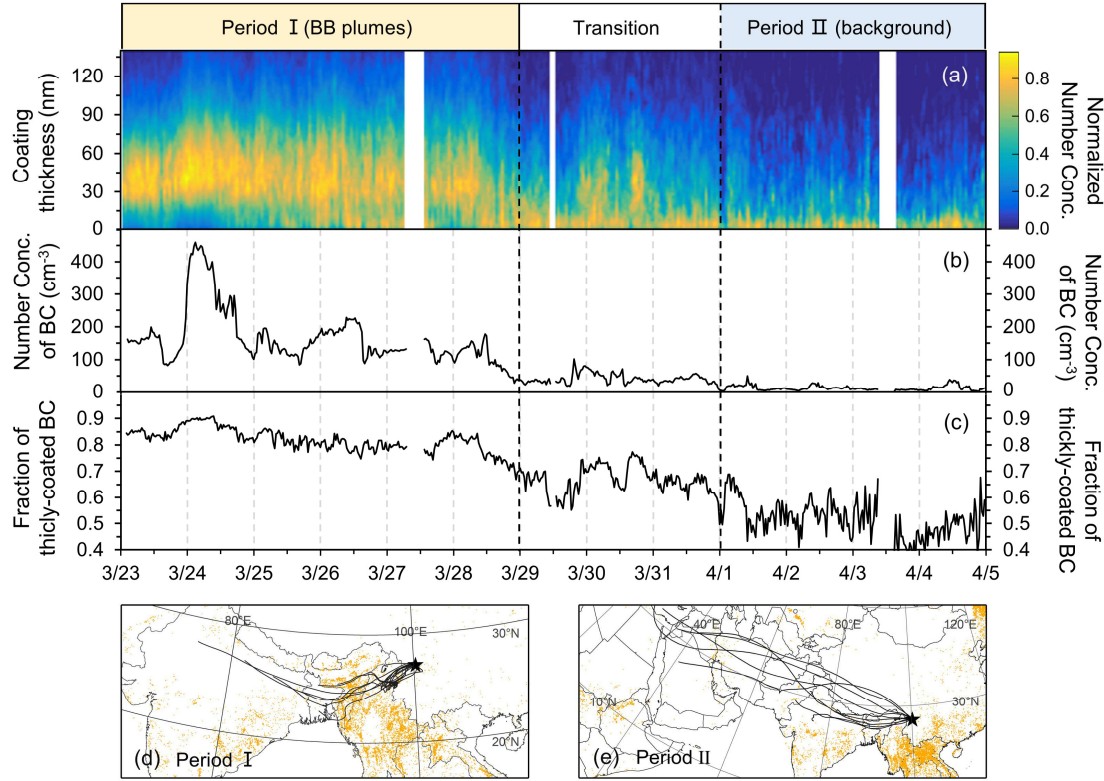

**Figure 2.** (a) Coating thickness and (b) number concentration of BC particles. (c) Number fraction of thickly-coated BC particles. 72 h back trajectories and active fire spots during (d) Period I and (f) Period II. The back trajectories were calculated by using HYSPLIT model (https://ready.arl.noaa.gov/HYSPLIT_traj.php). The active fire points were obtained from the Fire Information for Resource Management System (FIRMS), provided by the Moderate Resolution Imaging Spectroradiometer (MODIS) satellite (https://firms.modaps.eosdis.nasa.gov/map/).

Figure S4 shows the time series of aerosol mass concentration during this campaign. A pollution episode is marked out for the first few days (23 March to 29 March, marked as Period I), because they are characterized by high concentrations of all aerosol species. After that, the aerosol mass concentration gradually decreases and remains at an extremely low level in the last few days (1 April to 4 April, marked as Period II). In the following discussion, Period II will be regarded as the background





condition in this region. Then, it can be noticed from Fig. 2a, the coating thickness of BC particles shows a distinct difference between Period I and II: During Period I, the coating thickness is mainly distributed within 30-60 nm, but this mode disappears in Period II. The coating thickness in Period II is distributed around zero. It should be mentioned that the coating thickness here is only calculated for the particles with BC core diameter between 100-120 nm (mass-equivalent diameter). Aside from the coating thickness, the number fraction of thickly-coated BC also increases significantly during Period I, from an average

of 51.7% in Period II to 81.8% in Period I, with a maximum value of 90.8% (Fig. 1c). The discrepancy in the mixing state between the two periods adequately addresses that the BC particles observed during Period I are highly aged, while the BC particles observed during Period II are relatively fresh. Corresponding to this variation, the abundance of BC-containing particles also exhibits a large difference between Period I and II (Fig. 2b). The average BC number concentration for Period I is 165 cm$^{-3}$, with a maximum value of 460 cm$^{-3}$, whilst the average BC number concentration for Period II is only 13 cm$^{-3}$.

To further explore the origin of those abundant aged BC particles, the air mass back trajectories were investigated. During Period I, the air mass is mainly from the southwest and passes through the areas covered by dense fire spots (Fig. 2d). Differently, the trajectories during Period II are mainly from the west to northwest, with much longer trajectory length and in the absence of fire spots along the pathway. Hence, there is a strong possibility that, those abundant aged BC particles observed in Period I are originated from the fire smokes in South and Southeast Asia. This deduction is in accordance with the

conclusions given by Zheng et al. (2017) and Wang et al. (2019). Based on their chemistry-related evidence, those abundant aged BC are believed to have an origin of biomass burning.

The high concentration of BC particles in the pre-monsoon season over the TP is not an accidental phenomenon discovered only by our observation. Zhao et al. (2013), Cong et al. (2015a). and Sarkar et al. (2015) also found that the BC (or elemental carbon, EC) concentration reaches its maximum of a year in the pre-monsoon season in the Tibetan region, and they also

correlated this phenomenon to the transport of biomass burning (BB) aerosols in South Asia. However, our observation provides direct evidence that those BC particles are highly aged, indicating that those BC particles have experienced long-range transport to arrive on the TP. The occurrence of this kind of transport event in pre-monsoon season can be resulted from the prevalence of BB activities in South and Southeast Asia (Streets et al., 2003;Duncan et al., 2003) and the facilitation of the prevailing westerly wind during this period (Zhao et al., 2013;Cao et al., 2010). Since this kind of BB transport event commonly

occurs in a specific season every year, the researches on its impact on the TP area are of great importance.

## 3.2 Changes on the diurnal cycle of BC abundance and mixing state under the impact of transported BB plumes (implication for vertical distribution)

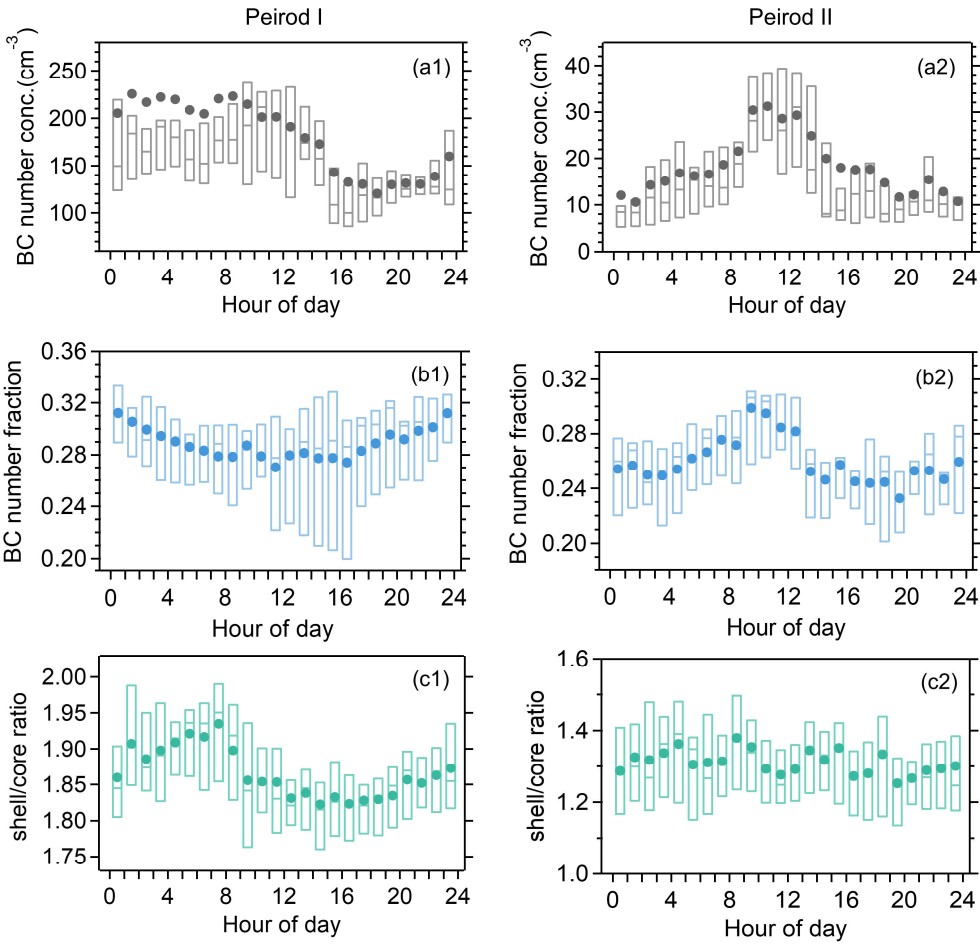

**Figure 3.** Diurnal variation of BC number concentration(a), BC number fraction (b) and shell/core ratio (c) during Period I and Period II.
The solid dots are the mean values. The rectangles cover the data from 25% to 75% quantile. The short lines in the middle of the rectangles represent the median values.

Vertical transport of surface air pollutants is an important source of air pollutants in high-altitude atmosphere. It can be reflected by the high pollutant concentration during the daytime due to the strong convective mixing. In a previous publication based on the same measurement, Shang et al. (2018) found an increasing trend of Aitken-mode particle number concentration during the daytime. By examining the available evidence, they believed that those Aitken-mode particles came from the primary emissions under the mountain. Here, our results show that the BC number concentration substantially increases (almost three times) during the daytime in Period II (Fig. 3a2), which coincides with Shang's findings. However, this diurnal pattern is changed in Period I. As shown in Fig. 3(a1), under the influence of transported BB plumes, the daytime increase of BC number concentration becomes less pronounced. This can be attributed to the comparable level of





BC concentration between the upper and surface atmosphere. Therefore, even though the vertical convection can bring the surface air up to the mountain top, it will not significantly elevate the BC number concentration in the upper atmosphere as the background case. This result indicates that the transported BB plumes will increase the BC concentration in the upper atmosphere, thereby reducing the difference in BC concentration between the upper and the surface atmosphere.

The diurnal cycle of BC relative abundance is also changed due to the transported BB plumes. In the background case of

Period II (Fig. 3(b2)), the number fraction of BC-containing particles shows an increasing trend during the daytime. A similar pattern was also found by Liu et al. (2020) at a mountain site located in the North China Plain, in which they used the mass fraction instead of the number fraction. Liu et al. believed that the increase of the relative abundance of BC during the daytime was owing to more efficient vertical transport of BC than other particle species. Besides, they also suggested that higher BC fraction led to a lower single scattering albedo, which resulted in an increase of positive radiative forcing.

However, this upward trend of BC number fraction during the daytime is almost reversed during Period I. As shown in Fig. 3(b1), the BC number fraction exhibits a slightly decreasing trend during the daytime, indicating a larger BC fraction in the upper air than the surface air. Therefore, when the vertical convection lofts the surface air to a higher altitude and mixes with the transported BB plumes, it will reduce the proportion of BC particles in the upper atmosphere.

Beside the absolute and relative abundance, the mixing state of BC is also essential to evaluate the radiative effects of BC.

Here, we use the mean shell/core ratio of BC particles with 100-120 nm core diameter (mass-equivalent diameter) to indicate the overall aging degree. A noticeable difference of the diurnal pattern of BC mixing state is also found between Period I and II. For the background condition (Period II), the shell/core ratios fluctuate at a lower level and do not show an obvious pattern (Fig. 3(c2)). This implicates that, in the background air, there is no significant difference in the aging degree between the BC particles at high altitude and near the ground. However, a clear drop of the shell/core ratios is observed during the

daytime of Period I (Fig. 3(c1)). Based on the above discussion, this drop can be caused by the mixing with the less aged BC, which were emitted from the near-ground sources and uplifted by the daytime convection.

Briefly, by comparing the different diurnal patterns during Period I and II, we found that the regional transport of BB plumes significantly changed the vertical variation of BC abundance (both absolute and relative abundance) and mixing state in background condition, which may lead to a profound impact on the radiative balance in the region.





**3.3 Variation of BC size distributions under the impact of transported BB plumes**

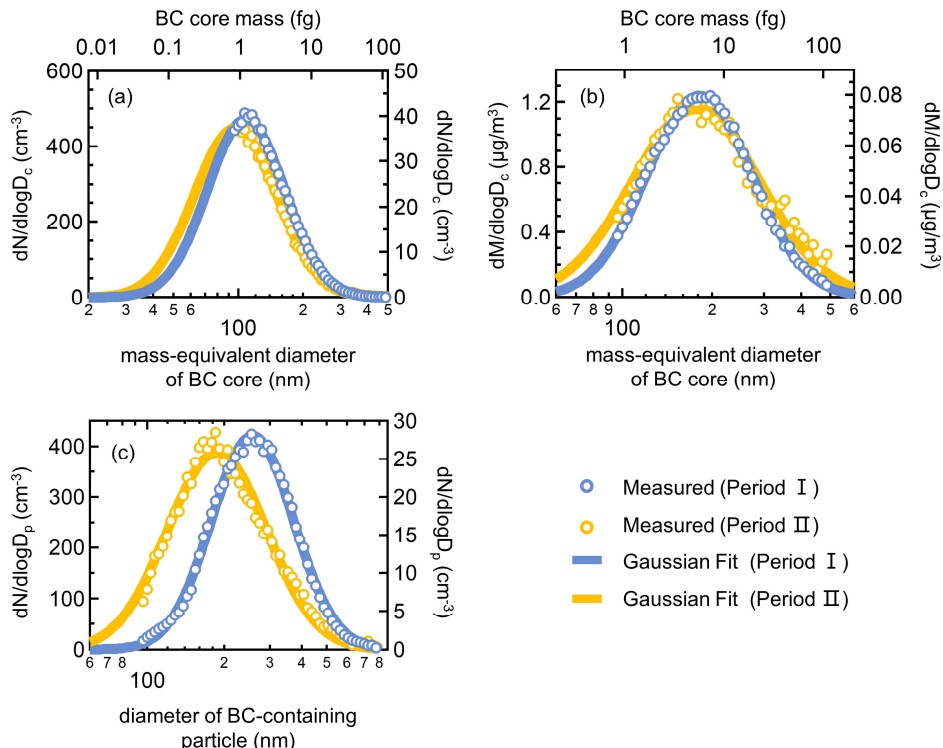

**Figure 4.** (a) Number and (b) mass size distribution of BC cores. (c) number size distribution of whole BC-containing particles. Left axis corresponds to the values in Period I, and right axis corresponds to the values in Period II. The mass-equivalent diameters of BC cores are obtained by assuming a density of 1.8 g/cm³.

The aging of BC particles is often caused by the secondary formation of coating materials or by the coagulation with co-
existing particles, both resulting in an increase of the particle size. Meanwhile, the BC mass in each BC-containing particle is
generally considered to remain unchanged during the aging process. Hence the size distribution of BC cores may give some
hints of BC's origin. Figure 4a shows the number size distribution of BC cores in Period I and II. A slightly larger size is
found in Period I. The fitted mean diameters for the two periods are 109 nm and 97 nm, respectively.  For the mass

distribution, the fitted mean diameters of each period are 181 nm and 177 nm, respectively (Fig. 4b). The difference of BC
core size distribution between these two periods confirms that the BC emission source in Period I and II could be different.
Additionally, the mean diameter of mass distribution observed in Period I is quite close to the values (181-182 nm) measured
from an aircraft flying over the northern India also in the pre-moon season (Brooks et al., 2019). This similarity further
verifies that those aged BC particles observed during Period I are possibly originated from the northern India, as the back

trajectories indicated (Fig. 2d). Figure 4c shows the number size distribution of whole BC particles (including the BC core
and non-BC coatings) during Period I and II. The fitted mean diameters of each period are 257 nm and 186 nm, respectively.
Since there is only a minor increase on the BC core size from the background air (Period II) to the BB plumes-influenced air



(Period I), the significant increase of the whole BC particle diameter is mainly resulted from the atmospheric aging, which are consistent with the discussion above (Section 3.1)

**3.4 Size-resolved mixing state of black carbon**

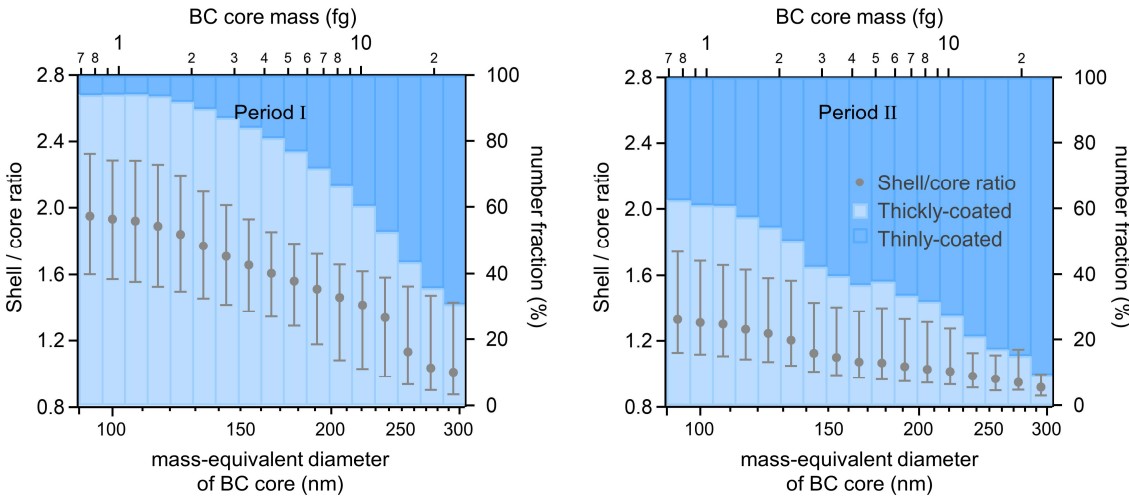

**Figure 5.** Size-resolved mixing state of black carbon during Period I and II

In previous sections, we demonstrate that BC particles of Period I have a higher aging degree than the background BC particles. However, the mixing state between single particles could be various, namely, different BC particles possibly have
a variety of aging degree. As illustrated in Fig. 5, the percentage of thickly-coated BC (light blue bars) decreases as the BC core size increases, from 94% at ~100 nm core diameter to 30% at ~300 nm core diameter in Period I. The median shell/core ratios in each core size bins also show a significant declining trend in Period I, from 1.95 dropping to 1.00. During Period II, although both the thickly-coated fractions and the shell/core ratios are much lower than those in Period I, they also show a decreasing trend as the BC core size increases. This implicates that the smaller BC cores are prone to obtain more coatings
than larger cores during the atmospheric aging process.

Nevertheless, this kind of heterogeneity in mixing state of BC is always neglected in most models, which can lead to an overestimate of the BC absorption enhancement if a uniformed shell/core ratio is assumed. Fierce et al. (2018) used a particle-resolved model to examine the influence of the heterogeneity in mixing state on the absorption enhancement, and found that the absorption is overestimated by as much as a factor of two if the diversity is neglected. In addition, their model-
based analysis also revealed that the coatings tend to be less on particles that contain large BC amounts, which is in line with our discovery above. Thus, it is essential to consider the heterogeneity of mixing state when modeling the optical properties of black carbon.





## 3.5 Strong enhancement of the light absorption due to the transported BB plumes

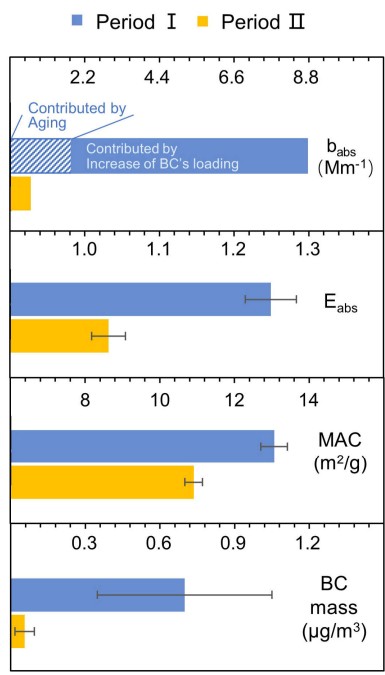

**Figure 6.** Comparison of BC mass, mass absorption cross-section, absorption enhancement and absorption coefficient in Period I and II (at the wavelength of 550 nm).

Given that the transported BB plumes significantly increase the BC loading and BC aging degree in the atmosphere, they will exert a large impact on the light absorption level of the atmosphere. Here, this impact is accordingly considered from two different aspects: the impact on the abundance and the impact on the mixing state. As shown in Fig. 6, the BC mass concentration is elevated by 13 times relative to the background air, indicating that the number of absorbing aerosols is substantially increased. Furthermore, due to the lensing effect, the thick coatings on those transported BC particles enhance the absorption ability of single BC particles, compared with the thinly-coated BC particles in the background air. Also as illustrated in Fig. 6, the mass absorption cross-section (MAC) is amplified by a factor of 1.2 relative to the background air, from 10.9 $m^2/g$ to 13.1 $m^2/g$. If compared to the uncoated BC particles, the enhancement factor ($E_{abs}$) will be up to 1.25 during Period I, while the enhancement factor of background BC particles is only 1.03. It also should be noticed that, the MAC and $E_{abs}$ values here reflect the absorption capacity of dry particles. If considering the uptake of water, the enhancement of light absorption will be more significant.

Combining these two aspects of the impact, the transported BB plumes eventually increase the total light absorption by a factor of nearly 15, from 0.6 $Mm^{-1}$ in Period II to 8.8 $Mm^{-1}$ in Period I (Fig. 6). In the total light absorption of Period I, 21% is contributed from the lensing effect which relate to the atmospheric aging during the long-range transport, and 79% is contributed from the increase of BC abundance. Particularly, the light absorption enhancement contributed by the aging have





far exceeded the background light absorption in Period II, emphasizing an important role of BC aging on the light absorption of the atmosphere in remote and clean areas. Meanwhile, the light absorption of the background air (Period II) is almost all contributed from the uncoated BC particles and the contribution from the lensing effect is neglectable.

In our previous publication, Wang et al. (2019) have showed that the light absorption at short wavelength by brown carbon is enhanced because of the transport of BB plumes during Period I. In addition to that research, this study reveals that the light absorption by black carbon, which can strongly absorb light at visible and infrared wavelength, is also strongly enhanced by the transported BB plumes. Overall, it can be concluded that the transported BB plumes occurring during the pre-monsoon season strongly enhance the light absorption of the atmosphere across the whole solar spectrum, which is bound to have a

profound radiative effect on the TP climate system.

## 4 Conclusions

Every year in the pre-monsoon season, the biomass burning plumes which originated in South and Southeast Asia can be transported to the Tibetan Plateau by the convenience of westerly wind. Based on field measurement, this study reveals the impact of the transported BB plumes on the abundance and mixing state of BC particles over the southeastern TP. As a result

of the transport of BB plumes, the number concentration of BC-containing particles was greatly increased, with the highest concentration exceeding the background air by more than 35 times. The percentage of thickly-coated BC was also increased significantly from 52% to as high as 91%. Those transported BC particles were found to have much thicker coatings and slightly larger core size than the background BC particles. The coating thickness varied among BC particles of different core sizes, showing a trend that the shell/core ratios decreased as the core size increased. Due to the thicker coatings, the mass

absorption cross-section of BC particles was enhanced by a factor of 1.2 compared to the background BC particles. Furthermore, the vertical variations of BC's abundance and mixing state were substantially changed because of the transport of BB plumes, resulting in a higher concentration, larger number fraction and higher aging degree of BC in the upper atmosphere compared to the background condition. The increase of both BC's abundance and aging degree eventually resulted in a 15-fold enhancement of the total light absorption on average. In the total light absorption, 79% was contributed by the

increase of BC's loading an 21% was contributed by the aging process during the long-range transport. Particularly, the light absorption enhancement induced by the BC aging has far exceeded the background light absorption. This study highlights the strong impact of the BC particles in the aged BB plumes on the TP climate system. The results in this study may be applicable to other regions beyond the TP, which is also influenced by aged BB plumes brought by long-range transport.

*Data availability.* The data presented in this article are available from the authors upon request (minhu@pku.edu.cn).



*Author contribution.* TT wrote the manuscript and analyzed the datasets. MH, GZ and ZW contributed important scientific thoughts and results' analysis. ZD, DS, JZ, YQ and ML performed the field experiment. YW, LZ and SG contributed to the field measurement.

*Competing interests.* The authors declare that they have no conflict of interest.

*Financial support.* This work was supported by the National Natural Science Foundation of China (91844301) and the China
Ministry of Environmental Protection Special Funds for Scientific Research on Public Welfare (201309016)

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
