# Peer review of "Strong Light Absorption Induced by Aged Biomass Burning Black Carbon over the Southeastern Tibetan Plateau in Pre-monsoon Season"

_Atmospheric Chemistry and Physics, 2020_

## Referee Comment (RC1) · Anonymous Referee #1 · 28 Dec 2020

This paper presents a field measurement at a remote background site to elucidate the influence of transported biomass burning (BB) plumes on the abundance and mixing state of BC particles over the southeastern TP. A combined system of differential mobility analyzer and single-particle soot photometer (DMA-SP2) was applied. Both the number concentration of BC-containing particles and the percentage of thickly-coated BC increased significantly during the pollution episode period due to the transport of BB plumes. Those transported BC particles were also found to have much thicker coatings and slightly larger core size than the background BC particles. Besides, the mass absorption cross-section and the total light absorption of BC particles also increased significantly, with 79

Specific comments:

Line 36: check the reference and clarify who first came up with the concept of "lensing effect".

Line 37: change to "requires the information of its size distribution..."

Line 51: Please add some more authoritative references rather than just "Zhao et al., 2013".

Line 72: change "import" to "transport"

Line 73: what's the criteria of the identification of pre-monsoon season?

Figure 1: I think there is a serious question for the current version of the map. The revised version need to correct the disputed boundary line between China and India.

Line 87: why not the typical 10:1 ratio between the sheath flow rate and the aerosol flow rate?

Line 92: Add the references. Equation 1: what's the diameters of Dp and Dc? Both the mobility diameters? Section 3.1: Although biomass burning was the major source of BC in the South Asia regions, it was not the only source for BC. For example, the fossil fuel combustions (vehicle exhausts and coal combustion) also emitted BC particles. So, it maybe not appropriate for the statement of "biomass burning black carbon" in your study.

Line 158: why calculated the coating thickness with BC core diameter only between 100-120 nm? How much of a deviation between it with the real ambient condition?

Line 159: it is better to do the comparison from Period I to Period II, namely decreased from Period I to Period II.

Line 166: it is strange why less fire spots in the Indo-Gangetic Plain where had serious pollution. Please check the fire data again.

Line 171: similar to above comment, the biomass burning is the major origin source for BC rather the only source.

Section 3.2: what's the reason for the large discrepancy between the mean and median values at 0-8h in Figure 3.2a1?

Line 192: As the author explained the high pollutant concentration occurred during the daytime due to the strong convective mixing, however, the highest BC number concentration in Figure 3.2a2 occurred at 10-13h, but showed obvious decrease in the afternoon when have the strong convective mixing conditions. Why? It is also inappropriate to say the number fraction of BC-containing particles shows an increasing trend during the daytime but decreased clearly after 13h.

Line 193-198: the explanation is not convincing. Firstly, the highest value was just 40 during the background P2 period, but larger than 200 during the P1 period, so it can not to say "This can be attributed to the comparable level of BC concentration between the upper and surface atmosphere". Secondly, the daily variation characteristics in Fig3a1 are not completely understood in this explanation, such as why the concentration is low during 20-24h but much higher during 0-8h? Overall, this section need to be rephrased clearly to explain why the diurnal patterns of BC number concentration are influence by the regional transport of BB plumes during the night but strong convective mixing during the daytime.

Line 231: " The difference of BC core size distribution between these two periods confirms that the BC emission source in Period I and II could be different." In my view, the size distribution in Period I and II are similar in Figure 4.

Line 234: it is insufficient to verify those aged BC particles are from the northern India just according to the close mean diameters. Rephrase the caption of Figure 5 clearly.

Line 249-250: "This implicates that the smaller BC cores are prone to obtain more coatings than larger cores during the atmospheric aging process" What's the reason?

[Figure]

Line 290: "As a result of the transport of BB plumes, the number concentration of BC-containing particles was greatly increased" It is not right to use the passive voice here in my opinion, please consider to change these in the entire conclusion section.
* * *

---

## Referee Comment (RC2) · Anonymous Referee #2 · 4 Jan 2021

This work observed size and mixing states of black carbon aerosols over a site on Yulong Mountain in southwestern China. With the monitoring data, authors calculated absorption of black carbon.

However, the observation duration is very short, about two weeks. Authors separated the duration to three time intervals. The background duration only covered 4 days. I don't understand why 4 days can represent a background, thus general conditions on the site.

Period I, which was defined as biomass burning event, covered 6 days. Comparing the 6-days data to the 4-days background, authors make results and conclusions. such

as; Resulted from both increase of BC loading and aging degree, the transported BB plumes eventually enhanced the total light absorption by 15 times, in which 21% was contributed by the BC aging and 79 % was contributed from the increase of BC mass.

With those very limited data output, authors claim this study revealed the impact of transported aged BB plumes on the atmospheric environment over the TP, which can serve as constraints for climate models to help with improving our understanding how human activities affect the global climate change.

I can not believe such short observation can have us a verified result.

Moreover, the biomass burning source identification is not well supported with tracer or receptor model methods, which should be typically applied for source identification.

My evaluation is that current observation is not enough verification to be published on ACP. Authors should extend observation for several repeats of events and longer background condition. Chemical analyzing should be useful for identification of sources before the definition of biomass events.

---

## Author Comment (AC1) · 9 Mar 2021

**Response to comment from Referee #1**

**This paper presents a field measurement at a remote background site to elucidate the state of BC particles over the southeastern TP. A combined system of differential mobility analyzer and single-particle soot photometer (DMA-SP2) was applied. Both the number concentration of BC-containing particles and the percentage of thickly-coated BC increased significantly during the pollution episode period due to the transport of BB plumes. Those transported BC particles were also found to have much thicker coatings and slightly larger core size than the background BC particles. Besides, the mass absorption cross-section and the total light absorption of BC particles also increased significantly.**

Response: We thank the reviewer for the careful reading of our manuscript and critical reviews as well as constructive suggestions. Here are our responses to the comments:

**1)  Line 36: check the reference and clarify who first came up with the concept of "lensing effect".**

Response: We thank the reviewer to point out this. We carefully checked the reference we quoted, *Impact of brown and clear carbon on light absorption enhancement, single scatter albedo and absorption wavelength dependence of black carbon (Lack and Cappa, 2010)*, however, we did not find the concept of "lensing effect" was quoted in this reference. Just in case, we also searched some related papers which mentioned the concept of "lensing effect", and we found none of them explained the concept as adequately as Lack and Cappa did. Therefore, we quoted this reference in our manuscript to introduce this concept of lensing effect.

**2)  Line 37: change to "requires the information of its size distribution..."**

Response: We thank the reviewer for the advice. We have changed it as suggested by the reviewer in our revised manuscript.

**3)  Line 51: Please add some more authoritative references rather than just "Zhao et al., 2013".**

Response: We thank the reviewer for the suggestion. Two more authoritative references relating to the facilitation of atmospheric circulation have been added in our revised manuscript:

Line 49-51: "With the facilitation of westerly atmospheric circulation during this period, the significant amounts of BC can be transported into the TP and become a major source of BC in Tibetan region (Bonasoni et al., 2010; Marinoni et al., 2010; Zhao et al., 2013)."

References:

Bonasoni, P., Laj, P., Marinoni, A., Sprenger, M., Angelini, F., Arduini, J., Bonafe, U., Calzolari, F., Colombo, T., Decesari, S., Di Biagio, C., di Sarra, A. G., Evangelisti, F., Duchi, R., Facchini, M. C., Fuzzi, S., Gobbi, G. P., Maione, M., Panday, A., Roccato, F., Sellegri, K., Venzac, H., Verza, G. P., Villani, P., Vuillermoz, E., and Cristofanelli, P.: Atmospheric Brown Clouds in the Himalayas: first two years of continuous observations at the Nepal Climate

Observatory-Pyramid (5079 m), Atmos Chem Phys, 10, 7515-7531, https://doi.org/10.5194/acp-10-7515-2010, 2010.

Marinoni, A., Cristofanelli, P., Laj, P., Duchi, R., Calzolari, F., Decesari, S., Sellegri, K., Vuillermoz, E., Verza, G. P., Villani, P., and Bonasoni, P.: Aerosol mass and black carbon concentrations, a two year record at NCO-P (5079 m, Southern Himalayas), Atmos Chem Phys, 10, 8551-8562, https://doi.org/10.5194/acp-10-8551-2010, 2010.

Zhao, Z. Z., Cao, J. J., Shen, Z. X., Xu, B. Q., Zhu, C. S., Chen, L. W. A., Su, X. L., Liu, S. X., Han, Y. M., Wang, G. H., and Ho, K. F.: Aerosol particles at a high-altitude site on the Southeast Tibetan Plateau, China: Implications for pollution transport from South Asia, J Geophys Res-Atmos, 118, 11360-11375, https://doi.org/10.1002/jgrd.50599, 2013.

**4) Line 72: change "import" to "transport"**

Response: We thank the reviewer for the suggestion. The word has been replaced as suggested by the reviewer in our revised manuscript. Please see the revised text below:

Line 71-72: "Different from the south edge of TP, the gentle slope of southeastern TP provides a geographical convenience for the transport of air pollutants from southern Asia to the TP."

**5) Line 73: what's the criteria of the identification of pre-monsoon season?**

Response: We thank the reviewer for raising the question. Here, we referred to the season identification proposed by Bonasoni et al. (2010, Table 1 and Figure 2) and Cong et al. (2015, Figure 2 and Table S1) in their papers. And these two references have been added in our revised manuscript. Instead of the Spring-Summer-Autumn-Winter season classification, the classification of Premonsoon-Monsoon-Postmonsoon-Winter seems more widely used in the researches on the Tibetan Plateau. Because the Tibetan Plateau is strongly influenced by the Indian monsoon system. In the above references, the criterion of the season identification is based on climatology, specifically including the relative humidity, temperature, wind direction and precipitation. In general, the monsoon season is characterized by high relative humidity, high temperature and frequent precipitation with prevailing southerly winds. In the remaining seasons, westerlies dominate the large-scale atmospheric circulation patterns with limited precipitation. The winter is featured by low relative humidity and low temperature. The pre-monsoon and post-monsoon are the transition period between the monsoon season and winter.

Line 72-73: "The measurement period lasted from 22 March to 4 April 2015, corresponding to the pre-monsoon season in the region (Bonasoni et al.,2010; Cong et al.,2015a)."

References:
Bonasoni, P., Laj, P., Marinoni, A., Sprenger, M., Angelini, F., Arduini, J., Bonafe, U., Calzolari, F., Colombo, T., Decesari, S., Di Biagio, C., di Sarra, A. G., Evangelisti, F., Duchi, R., Facchini, M. C., Fuzzi, S., Gobbi, G. P., Maione, M., Panday, A., Roccato, F., Sellegri, K., Venzac, H., Verza, G. P., Villani, P., Vuillermoz, E., and Cristofanelli, P.: Atmospheric Brown Clouds in the Himalayas: first two years of continuous observations at the Nepal Climate Observatory-Pyramid (5079 m), Atmos Chem Phys, 10, 7515-7531, https://doi.org/10.5194/acp-10-7515-2010, 2010.

Cong, Z., Kang, S., Kawamura, K., Liu, B., Wan, X., Wang, Z., Gao, S., and Fu, P.: Carbonaceous aerosols on the south edge of the Tibetan Plateau: concentrations, seasonality and sources, Atmos Chem Phys, 15, 1573-1584, https://doi.org/10.5194/acp-15-1573-2015, 2015a.

**6)    Figure 1: I think there is a serious question for the current version of the map. The revised version needs to correct the disputed boundary line between China and India.**

Response: We thank the reviewer for raising the question. To avoid controversy, we have removed all national boundaries in our revised manuscript. Please see the revised map below:

[Figure]

Figure 1 Location of measurement site at Mt. Yulong. Elevation data from SRTM 90m DEM Version 4, base map from ESRI. World Ocean Base, manipulating by ArcMap Version 10.2. Redlands, CA: ESRI, 2014.

**7)    Line 87: why not the typical 10:1 ratio between the sheath flow rate and the aerosol flow rate?**

Response: We thank the reviewer to bring attention to this question. We checked the original file of the DMA and found we did make a mistake on the flow rate of the sheath flow, which should be 3.0 L/min. We have corrected it in our revised manuscript. Please see the revised text:

Line 88: "the sheath flow rate of the DMA was 3.0 L/min".

**8)    Line 92: Add the references**

Response: We thank the reviewer for the suggestion. Two references have been added in our revised manuscript:

Line 92-93: "For non-absorbing particles, the peak intensity of the scattering signal is proportional to the scattering cross-section of the particle, which mainly depends on the particle size (Schwarz et al., 2006; Stephens et al., 2003)."

References:

Schwarz, J. P., Gao, R. S., Fahey, D. W., Thomson, D. S., Watts, L. A., Wilson, J. C., Reeves, J. M., Darbeheshti, M., Baumgardner, D. G., Kok, G. L., Chung, S. H., Schulz, M., Hendricks, J., Lauer, A., Karcher, B., Slowik, J. G., Rosenlof, K. H., Thompson, T. L., Langford, A. O., Loewenstein, M., and Aikin, K. C.: Single-particle measurements of midlatitude black carbon and light-scattering aerosols from the boundary layer to the lower stratosphere, J Geophys Res-Atmos, 111, https://doi.org/10.1029/2006JD007076, 2006.

Stephens, M., Turner, N., and Sandberg, J.: Particle identification by laser-induced incandescence in a solid-state laser cavity, Appl Optics, 42, 3726-3736, https://doi.org/10.1364/AO.42.003726, 2003

**9) Equation 1: what's the diameters of Dp and Dc? Both the mobility diameters?**

Response: Thank the reviewer to raise the question. As mentioned in Line 120-121, $D_p$ is the mobility diameter of the whole BC-containing particle, which was measured by the DMA (differential mobility analyzer) in our DMA-SP2 tandem system. As for $D_c$, it was calculated by dividing the BC core mass by the effective density of the BC core. The BC core mass in BC-containing particles was directly measured by the SP2(single-particle soot photometer), and the effective density of the BC core was optimized by matching the optical diameter retrieved from SP2 data and Mie calculation with the measured mobility diameter measured by DMA. More details on the retrieval of the effective density of BC core can be found in Zhang et al. (2018).

References:

Zhang, Y. X., Su, H., Ma, N., Li, G., Kecorius, S., Wang, Z. B., Hu, M., Zhu, T., He, K. B., Wiedensohler, A., Zhang, Q., and Cheng, Y. F.: Sizing of Ambient Particles From a Single-Particle Soot Photometer Measurement to Retrieve Mixing State of Black Carbon at a Regional Site of the North China Plain, J Geophys Res-Atmos, 123, 12778-12795, https://doi.org/10.1029/2018jd028810, 2018.

**10) Section 3.1: Although biomass burning was the major source of BC in the South Asia regions, it was not the only source for BC. For example, the fossil fuel combustions (vehicle exhausts and coal combustion) also emitted BC particles. So, it maybe not appropriate for the statement of "biomass burning black carbon" in your study.**

Response: We thank the reviewer for the suggestion. It is indeed not convincing enough to deduce that the aged BC observed in Period I is from the biomass burning (BB), merely based on the evidence of BC and fire spots. However, we believe those aged BC have an origin of biomass burning because it has been proved in two previously published papers (Wang et al., 2019; Zheng et al., 2017) by the chemical analysis. In these two references, both of which were based on the same field campaign with our study, Wang et al. and Zheng et al. reported that the main BB tracers, including the acetonitrile (Figure 5 in Zheng et al., 2017), levoglucosan (Figure 1 in Wang et al., 2019), potassium ion (K$^+$, Table S1 in Wang et al., 2019) and f$_{60}$ (Figure 2 in Zheng et al., 2017) had a much higher concentration level in Period I than those in Period II. Besides, Wang et al. also found that the concentration and the light absorption of HUmic-Like Substances (HULIS), major fractions of brown carbon also enhanced up to 42 times of the background levels in Period I. All of the above evidence suggests that the aged BC aerosols observed in Period I were highly likely from the biomass burning. Furthermore, besides our study, there are many other studies (Zhao et al., 2013;

Cong et al., 2015; Sarkar et al., 2015) that also correlated the high BC concentration observed in the pre-monsoon season to the transport of biomass burning aerosols in South Asia. For better understanding, we made a supplementary explanation in Line 170-173 for the source identification of the biomass burning, and also added a supplementary figure of BB tracers (Figure S5) in our revised manuscript. Please see the revised text below:

Line 145: "3.1 Observational evidence for the transport of black carbon aerosols" (removed the phrase "biomass burning")

Line 170-173: "This deduction is supported by the chemistry-related evidence provided by Zheng et al. (2017) and Wang et al. (2019). The concentrations of biomass burning tracers, including the acetonitrile, levoglucosan, and potassium ion are much higher in Period I than those in Period II (Figure S5). Hence, those abundant aged BC particles observed in Period I are believed to have a highly possible origin of biomass burning."

[Figure]

Figure S5. Temporal variation of the biomass burning tracers during the observation.

References:

Cong, Z., Kang, S., Kawamura, K., Liu, B., Wan, X., Wang, Z., Gao, S., and Fu, P.: Carbonaceous aerosols on the south edge of the Tibetan Plateau: concentrations, seasonality and sources, Atmos Chem Phys, 15, 1573-1584, https://doi.org/10.5194/acp-15-1573-2015, 2015a.

Sarkar, C., Chatterjee, A., Singh, A. K., Ghosh, S. K., and Raha, S.: Characterization of Black Carbon Aerosols over Darjeeling - A High Altitude Himalayan Station in Eastern India, Aerosol Air Qual Res, 15, 465-478, https://doi.org/10.4209/aaqr.2014.02.0028, 2015.

Wang, Y. J., Hu, M., Lin, P., Tan, T. Y., Li, M. R., Xu, N., Zheng, J., Du, Z. F., Qin, Y. H., Wu, Y. S., Lu, S. H., Song, Y., Wu, Z. J., Guo, S., Zeng, L. W., Huang, X. F., and He, L. Y.: Enhancement in Particulate Organic Nitrogen and Light Absorption of Humic-Like Substances over Tibetan Plateau Due to Long-Range Transported Biomass Burning Emissions, Environ Sci Technol, 53, 14222-14232, https://doi.org/10.1021/acs.est.9b06152, 2019.

Zhao, Z. Z., Cao, J. J., Shen, Z. X., Xu, B. Q., Zhu, C. S., Chen, L. W. A., Su, X. L., Liu, S. X., Han, Y. M., Wang, G. H., and Ho, K. F.: Aerosol particles at a high-altitude site on the Southeast Tibetan Plateau, China: Implications

for pollution transport from South Asia, J Geophys Res-Atmos, 118, 11360-11375, https://doi.org/10.1002/jgrd.50599, 2013.

Zheng, J., Hu, M., Du, Z. F., Shang, D. J., Gong, Z. H., Qin, Y. H., Fang, J. Y., Gu, F. T., Li, M. R., Peng, J. F., Li, J., Zhang, Y. Q., Huang, X. F., He, L. Y., Wu, Y. S., and Guo, S.: Influence of biomass burning from South Asia at a high-altitude mountain receptor site in China, Atmos Chem Phys, 17, 6853-6864, https://doi.org/10.5194/acp-17-6853-2017, 2017.

**11) Line 158: why calculated the coating thickness with BC core diameter only between 100-120 nm? How much of a deviation between it with the real ambient condition?**

Response: We thank the reviewer for raising the question. There are three reasons for choosing the coating thickness with BC core diameter between 100-120 nm. Firstly, the ambient BC particles are composed of BC particles of different sizes. And there is a difference in the coating thickness on particles of different sizes. Meanwhile, within the same particle size range, the coating thickness also varies between different BC particles (as shown in the contour plot in Figure R1 below). Thus, if the researchers intend to explore the temporal variation of BC coating thickness, there is a need to fix a size range. Secondly, during the aging process in ambient air, the whole particle size ($D_p$) of BC particles will change as the aging time increases. However, the BC core mass, which is contained in a single BC paritlce, is supposed to be unchanged during the aging process. Thus, the mass-equivalent BC core diameter (BC core mass divided by a fixed density of 1.8 g/cm$^3$) was used here. Thirdly, more particles in the range of 100-120 nm can ensure a solid calculation result. From the particle number size distribution (Figure R1 below, also the Figure 4a in our manuscript), it can be noticed that, the peak value was shown in the range 100-120 nm. More data for the calculation can minimize the interference of outliers as much as possible. Therefore, we used the coating thickness with mass-equivalent BC core diameter between 100-120 nm to explore the temporal evolution of BC aging degree in the ambient environment. Figure R1 shows the average coating thickness distribution in the real ambient condition of our observation. And the range of 100-120 nm is marked by grey area.

[Figure]

Figure R1. Average coating thickness distribution among different BC particles with varied mass-equivalent BC core diameters. The circles are the averaged particle number size distribution. The

area enclosed by the dotted line represents the 100-120 nm size range.

**12) Line 159: It is better to do the comparison from Period I to Period II, namely decreased from Period I to Period II.**

Response: We thank the reviewer for the suggestion. We have rewritten this sentence in our revised manuscript. Please see the revised text below.

Line 159-161: "Aside from the coating thickness, the number fraction of thickly-coated BC is also much higher in Period I than in Period II, with an average of 81.8 % in Period I compared with an average of 51.7% in Period II (Fig. 1c)."

**13) Line 166: it is strange why less fire spots in the Indo-Gangetic Plain where had serious pollution. Please check the fire data again.**

Response: We thank the reviewer for raising the question. We checked the fire data again and did not find any mistakes in it. From Figure 2d and 2e, it seems like the fire spots are less in Period I than in Period II. However, it should be noticed that the graphic scales of these two maps are different. Figure 2e (Period II, no pollution) covers larger area than the Figure 2d (Period I, serious pollution). If these two figures share same scales, as shown in Figure R2 below, the number of fire spots do not show large difference between the two periods. More importantly, whether the fire spots are present on the pathways of trajectories or not is more critical than the number of fire spots. During Period I, the trajectories are mainly from the southwest and pass through the areas covered by dense fire spots. However, the trajectories during Period II are mainly from the west and in the absence of fire spots along the pathway. The absence of fire spots on the trajectories means that the fire smoke is not likely to be transported to our observation site and lead to an air pollution. On the contrary, there are dense fire spots distributed around the trajectories in Period I, demonstrating that the pollution in Period I is possibly resulted from the transport of the fire smoke. Related analysis of the firs spots during our observation can also be found in Wang et al. (2019)'s paper.

[Figure]

[Figure]

Figure R2 Back trajectories (black lines) and active fire spots (orange dots) during Period I and Period II on the same graphic scale. The back trajectories were calculated by using HYSPLIT model (https://ready.arl.noaa.gov/HYSPLIT_traj.php). The active fire points were obtained from the Fire Information for Resource Management System (FIRMS), provided by the Moderate Resolution Imaging Spectroradiometer (MODIS) satellite (https://firms.modaps.eosdis.nasa.gov/map/).

Reference:

Wang, Y. J., Hu, M., Lin, P., Tan, T. Y., Li, M. R., Xu, N., Zheng, J., Du, Z. F., Qin, Y. H., Wu, Y. S., Lu, S. H., Song, Y., Wu, Z. J., Guo, S., Zeng, L. W., Huang, X. F., and He, L. Y.: Enhancement in Particulate Organic Nitrogen and Light Absorption of Humic-Like Substances over Tibetan Plateau Due to Long-Range Transported Biomass Burning Emissions, Environ Sci Technol, 53, 14222-14232, https://doi.org/10.1021/acs.est.9b06152, 2019.

**14) Line 171: similar to above comment, the biomass burning is the major origin source for BC rather the only source.**

Response: We thank the reviewer for the comment. The response for this comment can be found in Response (10) above. In our revised manuscript, we added a more detailed explanation and provided a supplementary figure (Figure S5) to support the source identification of the biomass burning.

**15) Section 3.2: what's the reason for the large discrepancy between the mean and median values at 0-8h in Figure 3.2a1?**

Response: We thank the reviewer for raising this question. The reason for the large discrepancy between the mean and median values during 0-8 h in Figure 3(a1, BC number concentration in Period I) is the extremely high concentration occurring during 0-8h on March 24[th]. As shown in Figure 2b of our manuscript, the BC number concentration reaches the maximum during 0-8h on March 24[th], which are much higher than the concentrations in the same time period of other days. The mean and median values shown in Figure 3(a1) are calculated based the data covering the period from March 23[rd] to 29[th]. Therefore, an extreme number from one day will increase the average level of all days; meanwhile, the median value is less affected by the extreme values.

**16) Line 192: As the author explained the high pollutant concentration occurred during the daytime due to the strong convective mixing, however, the highest BC number concentration in Figure 3.2a2 occurred at 10-13h, but showed obvious decrease in the afternoon when have the strong convective mixing conditions. Why? It is also inappropriate to say the number fraction of BC-containing particles shows an increasing trend during the daytime but decreased clearly after 13h.**

Response: We thank the reviewer for the comment. We attributed the daytime increase of the BC number concentration to the strong convective mixing because of the following two reasons. Firstly, there was no detected emission sources around our observation site. Thus, the increase of BC concentration is caused by the transport from other places. Secondly, we were inspired by previous studies (Liu et al., 2020; Lugauer et al., 2000; Nyeki et al.,1998; Shang et al., 2018), in which they also observed an increase of the particle concentration during the daytime at high-altitude sites, and

they also attributed the daytime increase to the thermally driven vertical transport. So, we deduced that the daytime increase of the BC number concentration was probably caused by the vertical transport. To further prove this speculation, we added an analysis of diurnal variation of the water vapor concentration in our revised manuscript. Water is expected to enter the air mostly by evaporation from the surface (Kivekäs et al., 2009). Here we calculated the water vapor concentration based on the temperature and relative humidity. As shown in Figure R3 below, the water vapor concentration also exhibits an increasing trend during 8-12 h, which coincides with the increase of the BC concentration. This synchronicity supports that the observation site was possibly influenced by the vertical transport of the air from the lower elevations. However, as the reviewer questioned, the water vapor concentration also decreases in the afternoon as the BC concentration does. This is likely due to the scavenging effect of the high-speed wind in the afternoon. As shown in Figure R3(b) below, the wind speed increases rapidly after 13 h. As for the inappropriate description of the trend of the BC number fraction, we rephrased the sentence in our revised manuscript. Please see the revised text below:

Line 193-199: "Here, our results show that the BC number concentration substantially increases (almost three times) during the period from 08:00 to 12:00 (local time) in Period II (Fig. 3(a2)), which coincides with the time when the water vapor concentration increases (Fig. S6(a2)). This indicates that the observation site was probably influenced by the vertical transport during this time. Because the water vapor in the atmosphere is supposed to mostly come from the evaporation of the surface (Kivekäs et al., 2009). Here, the water vapor concentration is calculated based on the temperature and relative humidity. After 13:00, both the BC and water vapor concentration show a decreasing trend, which could be resulted from the scavenging effect of the high-speed wind (Fig. S6(b2))."

[Figure]

Figure R3. Diurnal variation of absolute humidity (water vapor concentration) and wind speed in Period II

References:

Kivekäs, N., Sun, J., Zhan, M., Kerminen, V. M., Hyvärinen, A., Komppula, M., Viisanen, Y., Hong, N., Zhang, Y.,

Kulmala, M., Zhang, X. C., Deli-Geer, and Lihavainen, H.: Long term particle size distribution measurements at Mount Waliguan, a high-altitude site in inland China, Atmos Chem Phys, 9, 5461-5474, https://doi.org/10.5194/acp-9-5461-2009, 2009.

Liu, D. T., Hu, K., Zhao, D. L., Ding, S., Wu, Y. F., Zhou, C., Yu, C. J., Tian, P., Liu, Q., Bi, K., Wu, Y. Z., Hu, B., Ji, D. S., Kong, S. F., Ouyang, B., He, H., Huang, M. Y., and Ding, D. P.: Efficient Vertical Transport of Black Carbon in the Planetary Boundary Layer, Geophys Res Lett, 47, https://doi.org/10.1029/2020GL088858, 2020.

Lugauer, M., Baltensperger, U., Furger, M., Gaggeler, H. W., Jost, D. T., Nyeki, S., and Schwikowski, M.: Influences of vertical transport and scavenging on aerosol particle surface area and radon decay product concentrations at the Jungfraujoch (3454 m above sea level), J Geophys Res-Atmos, 105, 19869-19879, https://doi.org/10.1029/2000JD900184, 2000.

Nyeki, S., Li, F., Weingartner, E., Streit, N., Colbeck, I., Gaggeler, H. W., and Baltensperger, U.: The background aerosol size distribution in the free troposphere: An analysis of the annual cycle at a high-alpine site, J Geophys Res-Atmos, 103, 31749-31761, https://doi.org/10.1029/1998JD200029, 1998.

Shang, D. J., Hu, M., Zheng, J., Qin, Y. H., Du, Z. F., Li, M. R., Fang, J. Y., Peng, J. F., Wu, Y. S., Lu, S. H., and Guo, S.: Particle number size distribution and new particle formation under the influence of biomass burning at a high altitude background site at Mt. Yulong (3410 m), China, Atmos Chem Phys, 18, 15687-15703, https://doi.org/10.5194/acp-18-15687-2018, 2018.

**17) Line 193-198: the explanation is not convincing. Firstly, the highest value was just 40 during the background P2 period, but larger than 200 during the P1 period, so it can not to say "This can be attributed to the comparable level of BC concentration between the upper and surface atmosphere". Secondly, the daily variation characteristics in Fig3a1 are not completely understood in this explanation, such as why the concentration is low during 20-24h but much higher during 0-8h? Overall, this section need to be rephrased clearly to explain why the diurnal patterns of BC number concentration are influence by the regional transport of BB plumes during the night but strong convective mixing during the daytime.**

Response: We thank the reviewer for the comment, which reminds us that our explanation is not clear enough for the diurnal variation shown in Figure 3 of our manuscript, also reminds us that there is a need to make an additional analysis for the diurnal variation. We have rephrased this section as the reviewer suggested. For the questions raised by the reviewer above, we firstly need to make it clear that the influence of the regional transport of BB plumes not just happened during the night, but lasted for the whole time of Period I. The aim of our analysis is to show that how the diurnal pattern would be changed under the influence of the regional transport of BB plumes.

For the first question, the reason for the difference of the highest BC concentrations between Period I and II is the regional transport of BB plumes. Not only the upper atmosphere but also the lower atmosphere was influenced by the transport of air pollution, which resulted in a significant increase of the particle concentration both in the upper and lower atmosphere. Therefore, when the daytime convective mixing uplifted the particles in the lower atmosphere into higher atmosphere, it would lead to a much higher particle concentration in Period I (Figure 3a1) than Period II (Figure 3a2).

For the second question, a lower level of BC concentration during 20-24 h (actually 15-24 h) than 0-8 h in Figure 3a1 (Period I) might associate with wind scavenging as we discussed in Response 16. As shown in Figure S6 below (we have added this supplementary figure to our revised supplement file), there is an obvious lift of the wind speed at 15 h and the wind speed is higher

during 15-24 h than that during 0-8 h. Also, as the reviewer suggested, we have rephrased this section as the reviewer suggested. Please see our revised manuscript.

[Figure]

Figure S6 Diurnal variation of absolute humidity (water vapor concentration) and wind speed during Period I and Period II. The solid dots are the mean values. The rectangles cover the data from 25% to 75% quantile. The short lines in the middle of the rectangles represent the median values.

**18) Line 231: " The difference of BC core size distribution between these two periods confirms that the BC emission source in Period I and II could be different." In my view, the size distribution in Period I and II are similar in Figure 4.**

Response: We thank the reviewer to point out this. Although the difference is small, there is indeed a little difference between the mean diameters of the BC core size distribution in Period I and Period II. As illustrated in our manuscript, the fitted mean diameters of the number size distribution in Period I and Period II are 109 nm and 97 nm, respectively; and the fitted mean diameters of the number size distribution are 181 nm and 177 nm, respectively. The values of Period I are larger than those of Period II. We agree that this small difference of the mean diameters is insufficient to prove the difference of the BC source. Therefore, we have removed this sentence in our revised manuscript.

**19) Line 234: It is insufficient to verify those aged BC particles are from the northern India just according to the close mean diameters.**

Response: We agree with the reviewer and have removed the sentence in our revised manuscript.

**20) Rephrase the caption of Figure 5 clearly.**

Response: We thank the reviewer for the suggestion. We have rephrased the caption of Figure 5 as

following:

Line 244-245: "Figure 5. Variations of the number fraction of thickly-coated BC and the shell/core ratio among the particles containing varying BC mass during Period I and Period II."

**21) Line 249-250: "This implicates that the smaller BC cores are prone to obtain more coatings than larger cores during the atmospheric aging process" What's the reason?**

Response: We thank the reviewer for raising the question, which reminds us that the expression of this sentence is ambiguous. We have removed this sentence in our revised manuscript. The reason for the different shell/core ratios between smaller and larger BC cores is that the surface area of the smaller particles is smaller. Therefore, when the coating material with the same mass is added to their surfaces, the size of the smaller particles will increase more than the larger particles.

**22) Line 290: "As a result of the transport of BB plumes, the number concentration of BC-containing particles was greatly increased" It is not right to use the passive voice here in my opinion, please consider to change these in the entire conclusion section.**

Response: We thank the reviewer for the suggestion. The improper use of the passive voice has been changed in our revised manuscript.

Line 291: "As a result of the transport of BB plumes, the number concentration of BC-containing particles greatly increased"
Line 292-293: "The percentage of thickly-coated BC also increased significantly"
Line 297-298: "the vertical variations of BC's abundance and mixing state substantially changed under the impact of transported BB plumes"

---

## Author Comment (AC2) · 9 Mar 2021

**Response to comment from Referee #2**

**This work observed size and mixing states of black carbon aerosols over a site on Yulong Mountain in southwestern China. With the monitoring data, authors calculated absorption of black carbon. However, the observation duration is very short, about two weeks. Authors separated the duration to three time intervals. The background duration only covered 4 days. I don't understand why 4 days can represent a background, thus general conditions on the site.**

Response: Thank the referee for the comment. First of all, we want to clarify that, the time separation for Period II (background) is not aimed to represent the general condition on the site and we did not make such a similar statement in our manuscript. Our purpose of defining this period is to compare with Period I, thus to compare the condition unaffected by the transported BB plumes (Period II) and the condition affected by the transported BB plumes (Period I).

Secondly, we want to demonstrate that, the study of this manuscript is one of the pieces of an integrated and systematic atmospheric observation research. The time period (4/1/2015-4/4/2015) which we marked as Period II has been previously defined as the same background period in a few published papers. According to the papers of Zheng et al. (2017), Shang et al. (2018), Wang et al. (2019), they also marked the same time period (4/1/2015-4/4/2015) as the background period to illustrate the impact of transported BB plumes on the atmospheric environment of the site.

References:

Zheng, J., Hu, M., Du, Z. F., Shang, D. J., Gong, Z. H., Qin, Y. H., Fang, J. Y., Gu, F. T., Li, M. R., Peng, J. F., Li, J., Zhang, Y. Q., Huang, X. F., He, L. Y., Wu, Y. S., and Guo, S.: Influence of biomass burning from South Asia at a high-altitude mountain receptor site in China, Atmos Chem Phys, 17, 6853-6864, https://doi.org/10.5194/acp-17-6853-2017, 2017.

Shang, D. J., Hu, M., Zheng, J., Qin, Y. H., Du, Z. F., Li, M. R., Fang, J. Y., Peng, J. F., Wu, Y. S., Lu, S. H., and Guo, S.: Particle number size distribution and new particle formation under the influence of biomass burning at a high altitude background site at Mt. Yulong (3410 m), China, Atmos Chem Phys, 18, 15687-15703, https://doi.org/10.5194/acp-18-15687-2018, 2018.

Wang, Y. J., Hu, M., Lin, P., Tan, T. Y., Li, M. R., Xu, N., Zheng, J., Du, Z. F., Qin, Y. H., Wu, Y. S., Lu, S. H., Song, Y., Wu, Z. J., Guo, S., Zeng, L. W., Huang, X. F., and He, L. Y.: Enhancement in Particulate Organic Nitrogen and Light Absorption of Humic-Like Substances over Tibetan Plateau Due to Long-Range Transported Biomass Burning Emissions, Environ Sci Technol, 53, 14222-14232, https://doi.org/10.1021/acs.est.9b06152, 2019.

**Period I, which was defined as biomass burning event, covered 6 days. Comparing the 6-days data to the 4-days background, authors make results and conclusions. such as; Resulted from both increase of BC loading and aging degree, the transported BB plumes eventually enhanced the total light absorption by 15 times, in which 21% was contributed by the BC aging and 79 % was contributed from the increase of BC mass. With those very limited data output, authors claim this study revealed the impact of transported aged BB plumes on the atmospheric environment over the TP, which can serve as constraints for climate models to help with improving our understanding how human activities affect the global climate change.**

**I can not believe such short observation can have us a verified result.**

Response: We thank the referee for the comment. As mentioned above, the results presented in this manuscript were part of an integrated and systematical observation campaign. The biomass burning event in Period I and the definition of the background period have been verified by three published papers (Zheng et al., 2017; Shang et al., 2018; Wang et al., 2019). Therefore, we believe that such research methods have been academically recognized and have certain scientific significance. In our study, we conducted accurate measurement of the BC mixing state by using the well-recognized instruments. And the data was precisely calculated which was explicitly described in our manuscript. More importantly than the absolute length of the time period, the complete pollution event disturbed by the transported biomass burning plumes has been captured and targeted analysis has been carried out. And we also provided the evidence of biomass burning tracers in the following response, which was also added to our revised manuscript.

Therefore, we are confident that the results presented in our manuscript are scientifically verified. But we still thank the referee for the scientific advice. For further field observations in the future, we will try to conduct a more in-depth and targeted discussion on the time scale to make a better optimization. If the reviewer meant that the implication of our study should be rephrased more specifically, please see the revised text below:

Line 64-66: "By probing into the microphysical properties of BC aerosols, this study revealed that the recurring pre-monsoonal BC concentration peaks in the southeastern TP could be resulted from regional transport of aged BC aerosols, which led to a strong light absorption over the region."

**Moreover, the biomass burning source identification is not well supported with tracer or receptor model methods, which should be typically applied for source identification.**

Response: Thank the referee for the comment. The source identification of biomass burning has been verified by the tracers in the papers of Zheng et al. (2017) and Wang et al. (2019), which was also explained in Line 170-173 of our manuscript. In our revised manuscript, we also added a supplementary figure (Figure S5) to give further explanation. As shown in the Figure S5 below, the concentrations of levoglucosan, potassium and acetonitrile, the three widely used BB tracers, are much higher during Period I (biomass burning-influenced period, 3/23-3/29) than those during Period II (background period, 4/1-4/4).

[Figure]

Figure S5. Temporal variation of the biomass burning tracers during the observation.

**My evaluation is that current observation is not enough verification to be published on ACP. Authors should extend observation for several repeats of events and longer background condition. Chemical analyzing should be useful for identification of sources before the definition of biomass events.**

Response: Thank the referee for the comment. As mentioned in the response above, the results presented in this manuscript were part of an integrated and systematical observation campaign. The biomass burning event in Period I and the definition of the background period have been verified by three published papers (Zheng et al., 2017; Shang et al., 2018; Wang et al., 2019). Therefore, we believe that such research methods have been academically recognized and have certain scientific significance. More importantly than the absolute length of the time period, the complete pollution event disturbed by the transported biomass burning plumes has been captured and targeted analysis has been carried out.

By probing into the microphysical properties of BC aerosols, our study provides direct evidence that the recurring pre-monsoonal BC peaks in the Himalayas and Tibetan Plateau (HTP) region can result from the regional transport. Because these BC particles have a higher aging degree than the background BC particles. Before our study, there are many papers reporting that the BC concentration has a maximum value in pre-monsoon season around the year in HTP region (Decesari et al., 2010; Zhao et al., 2013; Marinoni et al., 2013; Cong et al., 2015; Marinoni et al., 2010). However, due to lack of information on the BC mixing state, they could only deduce from the back trajectories that these BC particles are possibly transported from the South Asia. By investigating the microphysical properties of BC particles, our study revealed that those transported BC particles are mostly thickly coated and have thicker coatings than the background BC. Based on this, our research elucidates that the aging process during the long-range transport would strengthen the radiative heating of those transported BC aerosols. Furthermore, based on the analysis of the diurnal variation, our research indicated that the regional transport of BB plumes might increase the BC concentration in the free troposphere, which would exert a more profound effect on the regional

climate system. Overall, we believe our study can expand our knowledge on atmospheric aerosols over the HTP region and may help us understand the climate change occurring in this region. Additionally, our research is one of the few studies that conducted field observations in the free troposphere to focus on the BC mixing state. Therefore, we believe our studies is qualified to be published on ACP.

References:

Cong, Z., Kang, S., Kawamura, K., Liu, B., Wan, X., Wang, Z., Gao, S., and Fu, P.: Carbonaceous aerosols on the south edge of the Tibetan Plateau: concentrations, seasonality and sources, Atmos Chem Phys, 15, 1573-1584, https://doi.org/10.5194/acp-15-1573-2015, 2015.

Decesari, S., Facchini, M. C., Carbone, C., Giulianelli, L., Rinaldi, M., Finessi, E., Fuzzi, S., Marinoni, A., Cristofanelli, P., Duchi, R., Bonasoni, P., Vuillermoz, E., Cozic, J., Jaffrezo, J. L., and Laj, P.: Chemical composition of PM10 and PM1 at the high-altitude Himalayan station Nepal Climate Observatory-Pyramid (NCO-P) (5079 m a.s.l.), Atmos Chem Phys, 10, 4583-4596, https://doi.org/10.5194/acp-10-4583-2010, 2010.

Marinoni, A., Cristofanelli, P., Laj, P., Duchi, R., Calzolari, F., Decesari, S., Sellegri, K., Vuillermoz, E., Verza, G. P., Villani, P., and Bonasoni, P.: Aerosol mass and black carbon concentrations, a two year record at NCO-P (5079 m, Southern Himalayas), Atmos Chem Phys, 10, 8551-8562, https://doi.org/10.5194/acp-10-8551-2010, 2010.

Marinoni, A., Cristofanelli, P., Laj, P., Duchi, R., Putero, D., Calzolari, F., Landi, T. C., Vuillermoz, E., Maione, M., and Bonasoni, P.: High black carbon and ozone concentrations during pollution transport in the Himalayas: Five years of continuous observations at NCO-P global GAW station, J Environ Sci, 25, 1618-1625, https://doi.org/10.1016/S1001-0742(12)60242-3, 2013.

Zhao, Z. Z., Cao, J. J., Shen, Z. X., Xu, B. Q., Zhu, C. S., Chen, L. W. A., Su, X. L., Liu, S. X., Han, Y. M., Wang, G. H., and Ho, K. F.: Aerosol particles at a high-altitude site on the Southeast Tibetan Plateau, China: Implications for pollution transport from South Asia, J Geophys Res-Atmos, 118, 11360-11375, https://doi.org/10.1002/jgrd.50599, 2013.

---

## Author Response (AR2)

**Response to Editor Decision**

Response: We thank for the editor's careful consideration and kind advice. We agree to transfer our manuscript to "Measurement Report".